# Understanding *Burkholderia glumae* BGR1 Virulence through the Application of Toxoflavin-Degrading Enzyme, TxeA

**DOI:** 10.3390/plants12233934

**Published:** 2023-11-22

**Authors:** Namgyu Kim, Duyoung Lee, Sais-Beul Lee, Gah-Hyun Lim, Sang-Woo Kim, Tae-Jin Kim, Dong-Soo Park, Young-Su Seo

**Affiliations:** 1Department of Integrated Biological Science, Pusan National University, Busan 46241, Republic of Korea; titanic622@pusan.ac.kr (N.K.); dlendud2164@pusan.ac.kr (D.L.); ghlim16@pusan.ac.kr (G.-H.L.); kimsw@pusan.ac.kr (S.-W.K.); tjkim77@pusan.ac.kr (T.-J.K.); 2National Institute of Crop Science, Milyang 50424, Republic of Korea; pappler@korea.kr (S.-B.L.); parkds9709@korea.kr (D.-S.P.)

**Keywords:** toxoflavin, toxoflavin-degrading enzyme, TxeA, *Burkholderia glumae*, transgenic plants, resistance

## Abstract

Rice (*Oryzae sativa* cv. dongjin) is a cornerstone of global food security; however, *Burkholderia glumae* BGR1, which is responsible for bacterial panicle blight (BPB), threatens its productive output, with dire consequences for rice and other crops. BPB is primarily caused by toxoflavin, a potent phytotoxin that disrupts plant growth at various developmental stages. Therefore, understanding the mechanisms through which toxoflavin and BPB affect rice plants is critical. Toxoflavin biosynthesis in *B. glumae* BGR1 relies on the *toxABCDE* operon, with ToxA playing a central role. In response to this threat, our study explores a metagenome-derived toxoflavin-degrading enzyme, TxeA, as a potential defense mechanism against toxoflavin’s destructive impact. TxeA-induced degradation of toxoflavin represents a potential strategy to mitigate crop damage. We introduce a groundbreaking approach: engineering transgenic rice plants to produce toxoflavin-degrading enzymes. These genetically modified plants, armed with TxeA, hold significant potential for combating toxoflavin-related crop losses. However, removal of toxoflavin, a major virulence factor in *B. glumae* BGR1, does not completely inhibit virulence. This innovative perspective offers a new shift from pathogen eradication to leveraging transgenic plants’ power, offering a beacon of hope for crop protection and disease management. Our study offers insights into the intricate interplay between toxoflavin, BPB, and TxeA, providing a promising avenue to safeguard rice crops, ensure food security, and potentially enhance the resilience of various agricultural crops to *B. glumae* BGR1-induced diseases.

## 1. Introduction

Rice (*Oryzae sativa* cv. dongjin) is a staple food crop that sustains billions of people worldwide, making its health and productivity crucial for global food security. *Burkholderia glumae* BGR1, which is a Gram-negative bacterium responsible for bacterial panicle blight (BPB), poses a significant threat to this vital crop. This devastating disease affects rice cultivation, damaging panicles and ultimately jeopardizing crop yields of tomatoes, eggplants, hot peppers, and sesame [1,2]. *B. glumae* has several virulence factors, such as toxins, proteases, lipases, extracellular polysaccharides, bacterial motility, and bacterial secretion systems [1,3,4,5,6]. Toxoflavin is the main pathogenic factor of *B. glumae* BGR1, and it causes serious symptoms not only in the flowering stage of rice plants but also in the vegetative and seedling stages. Toxoflavin, a broad host range phytotoxin that causes chlorotic damage, is a secondary metabolite produced by *B. glumae* BGR1. It is known for its phytotoxic properties and can have a significant impact on rice growth [1,2]. The major virulence factor of *B. glumae* BGR1 is the production of toxoflavin, a phytotoxic compound that negatively influences plant growth and plays a significant role in disease development resulting in rice grain rot and wilt, or BPB [7]. Toxoflavin inhibits the growth of rice roots when it comes into contact with them. This inhibition occurs because toxoflavin disrupts normal cellular processes in the plant roots, including cell division and elongation. Toxoflavin performs functions that include active electron transfer between NADH and oxygen and the production of hydrogen peroxide, which bypasses the cytochrome system, to induce symptoms [8,9]. Understanding the relationship between toxoflavin and BPB in rice plants is crucial for comprehending *B. glumae* BGR1’s virulence.

In *B. glumae* BGR1, toxoflavin biosynthesis and transport genes are encoded by the *toxABCDE* and *toxFGHI* operons, respectively [10]. The *toxABCDE* operon responsible for toxoflavin biosynthesis is polycistronic and comprises five genes: *toxA*, *toxB*, *toxC*, *toxD*, and *toxE* [11]. ToxA, an S-adenosyl-L-methionine-dependent methyltransferase, is predicted to be an N-methyltransferase that mediates the final step in the biosynthesis of toxoflavin and its congeners [12]. The *toxA* gene of the *toxABCDE* operon serves as a key intermediate in the toxoflavin biosynthesis pathway. The expression of genes belonging to the tox operon, including *toxA*, is regulated by the LysR-type regulator, ToxR, which is synthesized in part via a biosynthetic pathway common to the synthesis of riboflavin, starting with GTP as the precursor [7,10]. In addition, *toxABCDE* operons are controlled by quorum-sensing (QS) systems. QS is a cell-to-cell communication mechanism that regulates many physiological systems related to the virulence of *B. glumae* BGR1, including flagellar biosynthesis for bacterial motility [3,13].

Bacteria have evolved diverse defense mechanisms to counter toxins and ensure their survival in hostile environments. Some bacteria possess proteins or enzymes that degrade pathogenic toxins or prevent the toxic effects of other microorganisms. TxeA was the first metagenome-derived toxoflavin-degrading enzyme discovered by screening for enzyme activity that neutralizes the growth inhibition effect, based on a metagenomic DNA library of soil samples [14]. The gene encoding toxoflavin lyase (*tflA*) has been reported previously [15]. Unlike TfiA, TxeA is not only composed of a shorter length of 140 amino acid residues but also has a functionally superior toxoflavin decomposition function, with high activity over a broad temperature range and under acidic conditions [14].

Exploring toxoflavin degradation strategies in rice is vital for managing this pathogen and improving the health of rice crops. Quorum quenching is widely used as an antiviral strategy to alleviate or weaken the virulence of plant pathogenic bacteria. Lactonase from *Bacillus* spp., which hydrolyzes N-acyl homoserine lactone (AHL) to inactivate signaling molecules, has been used in transgenic plants to confer resistance to pathogenic bacteria [16]. Instead of solely focusing on pathogen eradication, we emphasize innovative strategies that leverage the power of transgenic plants. These genetically engineered plants, capable of producing enzymes that degrade toxoflavin, offer a potential solution to mitigate the adverse effects of this phytotoxin, thereby enhancing plant health and mitigating the detrimental effects of toxoflavin. Our novel approach holds promise for crop protection and disease management.

## 2. Results

### 2.1. Construction of Toxoflavin-Deficient Mutant Strains to Assess Whether Metagenome-Derived TxeA Is Functional in B. glumae BGR1 In Vivo

To evaluate whether the metagenome-derived toxoflavin-degrading enzyme TxeA actually maintains the functional ability to degrade toxoflavin in *B. glumae* BGR1 in vivo, its corresponding gene, *txeA*, was introduced into *B. glumae* BGR1 by conjugation. *B. glumae* BGR1, carrying the externally introduced *txeA*, was selected from kanamycin-containing media and confirmed by PCR. Mutant strains in which *txeA* was introduced externally, called TXE, were generated (Figure 1). The *toxA* marker-less mutant strain was constructed as a marker-less deletion mutant targeting the *toxA* gene *bglu_2g06400* via two homologous recombination events. After the completion of the second recombination, the *toxA* deletion mutant strain was confirmed by PCR (Figure 1). A single deletion mutant, called *ΔtoxA*, corresponding to the *toxA* gene, was generated. *ΔtoxA* was used to determine phenotypic comparisons related to toxoflavin production in TXE.

### 2.2. Phenotypic Analysis of Toxoflavin Production and Bacterial Motility in B. glumae BGR1 and Mutant Strains

The biggest characteristic of the toxoflavin produced by *B. glumae* BGR1 is that it has a yellow pigment that can be seen with the naked eye. After three days of culture of *B. glumae* BGR1 and mutant strains, the agar plates, including the peripheries of the colonies, were stained yellow because of the toxoflavin yellow pigment. Only *B. glumae* BGR1 exhibited the strongest yellow coloration (Figure 2A). In TXE, like in *ΔtoxA*, observable yellow pigment caused by toxoflavin was not observed on agar plates (Figure 2A). Based on previous studies, toxoflavins were detected by analyzing the compounds secreted by *B. glumae* BGR1 and other mutant strains using thin-layer chromatography [5,7,17]. Toxoflavin was detected only in *B. glumae* BGR1 under UV light at 365 nm. The toxoflavin-positive band in *B. glumae* BGR1 was not detected in other mutant strains containing *ΔtoxA* and TXE (Figure 2B). As a result, *ΔtoxA* and TXE indicate that they do not produce toxoflavin according to both simple visual inspection and UV light examination. These results also suggest that the introduced foreign gene encoding TxeA into *B. glumae* BGR1 was expressed and functional in *B. glumae* BGR1 in vivo. Bacterial motility was observed in all strains. None of the strains, including the *txeA*-expressing strain, showed motility defects (Figure 2C). In other words, no apparent difference in swarming motility was observed between the mutant strains and *B. glumae* BGR1; that is, comparable swarming motility was observed in all strains, as assessed by the formation of dendritic patterns.

### 2.3. Association between Toxoflavin and Virulence in a Mutant Strain That Does Not Produce Toxoflavin through an In Vivo Virulence Assay

To investigate whether the TXE strain completely degrades toxoflavin in vivo, thereby reducing its virulence by not producing toxoflavin, we performed an in vivo virulence assay at the following stages in rice plant development: Seed germination, vegetative, and flowering stages. For the seed germination assay, seed pre-germination was performed with a bacterial suspension, and the disease symptoms of growth reduction were observed seven days later (Figure 3C). The degree of disease symptoms at the seed germination stage was quantified using stem length (Figure 3E). When non-infected seeds germinated, the stem length was 4.96 ± 1.09 cm, and when *B. glumae* BGR1-infected seeds germinated, the stem length was 1.80 ± 0.32 cm. The stem length of *ΔtoxA*-infected seeds was 3.56 ± 0.33 cm. The stem length of seeds infected with the TXE strain was 3.92 ± 0.28 cm. Stem length, which indicates the degree of virulence at the seed germination stage, showed little growth reduction in seeds infected with TXE compared to those infected with *B. glumae* BGR1. Likewise, the virulence assay results for *ΔtoxA* showed a similar pattern to those obtained with the TXE strain. These virulence test results showing the attenuated virulence of *ΔtoxA* and TXE strains compared to *B. glumae* BGR1 were consistent with results obtained at the seed germination stage in both the vegetative stage and flowering stage (Figure 3A and Figure 4A). Specifically, disease severity (scale: 0–5) at the flowering stage was 2.71 ± 0.22 in *B. glumae* BGR1 and decreased to 2.71 ± 0.22 in TXE, to 3.03 ± 0.33 in *ΔtoxA* (Figure 3D). These results show that the insertion of an external *txeA* gene encoding a toxoflavin-degrading enzyme confers attenuated virulence by removing toxoflavin in vivo.

### 2.4. txeA-Induced Resistance against B. glumae BGR1 in Rice Plants

To investigate whether *txeA*-expressing rice plants were resistant to *B. glumae* BGR1, two homozygous lines (PIPK-6-14 and PIPK-10-8) were used. The resistance was assessed based on the disease severity of BPB in rice, and an in vivo virulence assay of BPB was performed as described above. The disease severity values of PIPK-6-14 and PIPK-10-8 were 3.14 ± 0.09 and 3.22 ± 0.28, respectively, which were lower than that for uninfected rice, dongjin, at 4.15 ± 0.18 (Figure 4A,B). Thus, the attenuated virulence of *B. glumae* BGR1 in *txeA*-expressing transgenic plants indicates that they are resistant to *B. glumae* BGR1.

## 3. Discussion

TxeA, a toxoflavin-degrading enzyme, was identified using metagenomic analysis [14]. The function and enzymatic activity of TxeA have been analyzed in terms of its biochemical characterization in vitro [14]. TxeA, which has a wide temperature range and high activity under acidic conditions, has a better ability to degrade toxoflavin than the previously reported TfiA [14,15]. Therefore, this study aimed to confirm the relationship between toxoflavin, a major virulence factor, and BPB caused by *B. glumae* BGR1, using TxeA. In particular, we confirmed whether the *txeA* gene was translated into a protein and was functional in an in vivo assay using rice plants showing toxoflavin symptoms; additionally, we confirmed its contribution to *B. glumae* BGR1 virulence. Our investigation involved the introduction of a metagenome-derived toxoflavin-degrading enzyme, the creation of a gene deletion mutant incapable of toxoflavin synthesis, and an in-depth analysis of its effects on bacterial virulence and plant resistance.

The introduction of foreign genes has mainly been attempted to induce the overexpression of target proteins or to compensate for deficient functions in a specific organism. For example, by introducing *tofI* and *tofR* genes into non-pathogenic *Burkholderia gladioli* KACC11889, which lacks the QS system that acts as an on/off switch for toxoflavin biosynthesis, the regulatory mechanism can be restored and virulence can be acquired by producing toxoflavin [15]. *toxA* is a major gene involved in toxoflavin synthesis, and *toxA* deletion mutant strains do not produce toxoflavin [7]. However, in order to effectively compare the toxoflavin-degrading enzyme, TxeA, with the TXE strain capable of producing it, the *ΔtoxA* strain was generated through marker-less deletion targeting the *toxA* gene of bglu_2g06400. TXE and *ΔtoxA* mutant strains failed to produce toxoflavin. *toxA* was deleted from the two mutant strains using various approaches, with the TXE mutant strain using a gain-of-function conceptual approach and *ΔtoxA* using a loss-of-function conceptual approach. Even though the approaches for removing toxoflavin were different, toxoflavin was not detected in the TXE mutant strain and *ΔtoxA* (Figure 2). In addition, the ToR in *B. glumae* BGR1 requires toxoflavin to induce the expression of two operons involved in toxoflavin synthesis and transport [7]. Therefore, toxoflavin degradation by TxeA in *B. glumae* BGR1 in vivo confers virulence with an attenuated gain-of-function. The absence of toxoflavin in *B. glumae* BGR1 was not observed with TxeA, suggesting that metagenome-derived TxeA fully performs its function in *B. glumae* BGR1 in vivo, decomposing only toxoflavin without any genome editing artifacts.

In vivo virulence assays on rice plants at different growth stages using *ΔtoxA* and TXE strains were performed to investigate the relationship between toxoflavin and virulence. We aimed to determine whether the TXE strain, which effectively degrades toxoflavin, exhibits reduced virulence compared to *B. glumae* BGR1 and the *ΔtoxA* mutant. The results reveal a compelling pattern. During seed germination, *B. glumae* BGR1 causes a substantial reduction in stem length, reflecting its high virulence. In contrast, both the *ΔtoxA* and TXE strains induced significantly milder symptoms, with less pronounced growth reduction (Figure 3C,E). These findings suggested that the absence of toxoflavin production, either through genetic deletion or enzymatic degradation, attenuated the virulence of *B. glumae* BGR1. Consistent with these observations, similar trends were observed at the vegetative and flowering stages of rice growth. The disease severity score indicated that both the *ΔtoxA* and TXE strains exhibited reduced virulence compared to *B. glumae* BGR1 (Figure 3B,E). This shows that loss-of-function *ΔtoxA* and gain-of-function TXE strains show a similar level of attenuation in virulence. Many studies have reported that toxoflavin from *B. glumae* BGR1 plays a major role in rice virulence [5,12,18,19,20]. However, in the in vivo virulence assay, the virulence of the *ΔtoxA* and TXE strains was not dramatically reduced or completely lost compared to the results of Kim et al. [7]. This is because, in addition to toxoflavin, other virulence factors of *B. glumae* BGR1 are present. Other pathogenic factors, such as bacterial motility, bacterial secretion systems, proteases, extracellular polysaccharides, and lipases, also play a greater or lesser role in the virulence of *B. glumae* BGR1 [3,4,5,6,10]. Indeed, the *ΔtoxA* and TXE strains only fail to produce toxoflavin, but bacterial motility still remains (Figure 2C). Toxoflavin synthesis and bacterial motility are regulated by an upstream regulator, the QS system of *tofI*/*tofR* [7,21]. The fact that the two mutant strains were not involved in bacterial motility does not mean that toxoflavin was not synthesized because of the disruption of the QS system. This indicated that the QS system was not destroyed or disordered during the failure of the two mutant strains to produce toxoflavin.

Furthermore, our research successfully created transgenic rice plants capable of expressing metagenome-derived TxeA. One of the most exciting and potentially transformative findings of this study is the enhanced resistance observed in plants that were introduced with the toxoflavin-degrading enzyme TxeA and subsequently infected with *B. glumae* BGR1 (Figure 4A,B). In other words, the *txeA* gene in the rice is translated within the plant to produce TxeA, a protein capable of degrading toxoflavin, which decomposes the toxoflavin produced by *B. glumae* BGR1, thereby removing one of the main pathogenic factors of *B. glumae* BGR1. These findings demonstrate the potential of TxeA to regulate toxoflavin production in *B. glumae* BGR1 and confer resistance in transgenic rice. In addition, the central role of toxoflavin in the ability of bacteria to infect rice plants has been highlighted. However, *txeA*-expressing transgenic plants did not completely eliminate the virulence of *B. glumae* BGR1, but instead showed attenuation of virulence. These results are consistent with the results of in vivo virulence assays performed at various rice growth stages with *ΔtoxA* and TXE mutants. This attenuated virulence may be attributed to virulence factors other than toxoflavin in *B. glumae* BGR1.

Future investigations should delve deeper into the regulatory mechanisms governing toxoflavin synthesis and its interactions with other virulence factors. The results of this study provide valuable insights into the role of toxoflavin and the potential application of a metagenome-derived toxoflavin-degrading enzyme, TxeA, to mitigate the virulence of *B. glumae* BGR1, particularly in rice. One example is the discovery of a new virulence factor in *B. glumae* BGR1, which is either weak or overshadowed by the main virulence factor. Additionally, pathogenic factors other than toxoflavin, which is the main cause of the disease caused by *B. glumae* BGR1, play a role in causing disease symptoms. By targeting the virulence factors of *B.glumae* BGR1 in a precise and environmentally friendly manner, we can pave the way for a more sustainable and resilient future for rice agriculture, ensuring food security while minimizing the ecological footprint of crop production. Our study represents a significant step toward achieving this goal and offers hope for a brighter future of rice agriculture.

## 4. Materials and Methods

### 4.1. Bacterial Strains Plasmids and Growth Conditions

All bacterial strains and plasmids used in this study are listed in Table 1. *B. glumae* BGR1 mutant strains and Escherichia coli were cultured on Luria-Bertani (LB) agar. A single colony grown on an LB agar plate was incubated in liquid medium in a shaking incubator at 200 rpm. When required, LB media were supplemented with antibiotics at the following concentrations: rifampicin, 100 μg/mL; kanamycin, 100 μg/mL; and phosphinothricin, 100 μg/mL.

### 4.2. Construction of B. glumae BGR1 Expressing the txeA Gene Encoding Toxoflavin-Degrading Enzyme (TxeA)

The sequence of the *txeA* gene, which encodes the toxoflavin-degrading enzyme (TxeA), was obtained from pQE::*txeA* [5]. Plasmid DNA was isolated using a Dokdo-PrepTM Plasmid DNA Mini-prep kit (Elpis Biotech, Daejeon, Republic of Korea). Polymerase chain reaction (PCR) was performed using SolgTM Pfu-X DNA polymerase (Solgent, Daejeon, Republic of Korea) to prepare fragments for cloning in pBBR1MCS2::*txeA* and pCAMBIA3301::*txeA*. The corresponding primer sets are listed in Table 2. The entire reading frame of the *txeA* gene was amplified by PCR using the primer sets pBBR_TxeA_cF/pBBR_TxeA_cR and pCAM_TxeA_cF/pCAM_TxeA_cF, and the amplified fragments were cloned into the broad host range expression vector pBBR1MCS2 and the binary vector for the T-DNA construct, pCAMBIA3301. The amplified fragments and vectors were digested with BamHI and EcoRI (NEB, Ipswich, MA, USA) and ligated. The cloned vectors were then conjugated to *B. glumae* BGR1 by co-culturing with *E. coli* S17-1. Mutant strains with cloned vectors containing *txeA* were selected on rifampicin (100 μg/mL) and kanamycin (60 μg/mL)-containing media and were confirmed by PCR using the primer sets pBBR_TxeA_cF/pBBR1MCS2_dR and pCAM_TxeA_cF/pCAM_TxeA_cR. All PCR was performed in 50 μL total volumes following the protocol. The PCR conditions were as follows: 2 min at 95 °C, followed by 35 cycles of 30 s at 95 °C, 30 s at the annealing temperature, and 30 s/kb at 72 °C. PCR was performed to verify the mutant strains generated as described above. PCR was performed using a SureCycler 8800 Thermal Cycler (Agilent Technologies, Santa Clara, CA, USA).

### 4.3. Generation of Marker-Less Deletion Mutant Strains in B. glumae BGR1

To construct marker-less deletion mutant strains, the plasmid pK18*toxA*, harboring portions of the *toxA* gene upstream (L fragment) and downstream (R fragment), was prepared. To prepare pK18toxA, the L and R fragments were amplified from the genomic DNA of *B. glumae* BGR1. The corresponding primer sets are listed in Table 2. The R fragment has a sequence that overlaps 15 bp of the 3′- end of the L fragment. The L and R fragments were mixed in a 1:1 ratio and amplified using toxA_LF and toxA_RR to create an overlap extension PCR product, which was a ligated LR fragment. The ligated LR fragment and pK18*mobsacB* were digested with EcoRI and HindIII and ligated. Competent *E. coli* DH5α cells were transformed with the recombinant plasmid, pK18*toxA*, and cells harboring pK18*toxA* were then selected in kanamycin (60 μg/mL)-containing media. pK18*toxA* was transformed into *E. coli* S17-1. pK18toxA was transformed into *B. glumae* BGR1 by conjugation via co-culture with *E. coli* S17-1 on LB agar plates. Then, the first crossover was selected on rifampicin (100 μg/mL) and kanamycin (60 μg/mL)-containing media. Three days later, the first crossover of colonies grown on the plate was selected and confirmed by PCR using the primer sets toxA_UP_F and pk18_DONW_R. The selected cells underwent a second crossover with two subcultures in LB broth every 12 h. The second crossover in the cells was selected on LB agar plates containing 30% sucrose and rifampicin (100 μg/mL). Finally, *toxA* marker-less mutant strains were confirmed by PCR using primer sets for toxA_UP_F and toxA_DOWN_R.

### 4.4. Observation and Detection of the Virulence Factors in B. glumae BGR1

Phenotypic characterization of the major pathogenic factors of *B. glumae* BGR1 was performed using thin-layer chromatography analysis of toxoflavin and bacterial motility at the appropriate infection site. To detect toxoflavin in *B. glumae* BGR1 and mutant strains, each strain was cultured in a 2 mL microcentrifuge tube containing 1.5 mL of LB broth at 37 °C for 1 day. The subsequent experiments were performed according to the method described by Lee et al. [5]. To evaluate bacterial motility, a bacterial swarming motility assay was performed on LB agar (0.5%, *w*/*v*) plates. The cultured cells (1 mL) were pelleted by centrifugation at 900× *g* for 2 min. The harvested cells were washed with 1 mL of fresh LB liquid medium, and the centrifugation step was repeated. The washed pellet was resuspended in 100 μL of distilled water, and 1 μL of the suspension was spotted onto the assay plate. The assay plate was incubated at 37 °C for 24 h.

### 4.5. In Vivo Virulence Assays at Various Developmental Stages of Rice Plants

To confirm the relationship between toxoflavin and virulence as a major virulence factor of *B. glumae* BGR1, and the functional aspects of TxeA, a toxoflavin-degrading enzyme, in vivo virulence assays were performed at various growth stages of rice plants: Seed germination, plant vegetation, and flowering. For the seed germination assay, rice seeds (*Oryza sativa* L. cv. dongjin) were pre-germinated at 30 °C in a 5 mL of bacterial suspension; when the optical density at 600 nm (OD_600_) reached 0.5, which is approximately 1.2 × 10^8^ colony-forming units (CFU)/mL, cultured bacterial cells in the mid-logarithmic phase were harvested via centrifugation, washed, and resuspended in distilled water. Two days later, the germinating seeds were transferred to a sterile plant culture dish (SPL Life Sciences, Pocheon, Republic of Korea) containing 20 mL of distilled water and were grown for 7 days at 30 °C under the following conditions: High humidity (close to 100%) in closed environments, 16 h of light provided by 400 V lamps, during the day; high humidity (close to 100%) in closed environments, 8 h of darkness, at night. The damage to and length of rice seeds were assessed 7 days after inoculation. To perform in vivo virulence assays in the vegetative and flowering stages, experimental soil was taken from the nearest rice field. Surface-sterilized, healthy rice seeds were selected for single-pot experiments. For in vivo virulence at the vegetative stage, the optical density of the bacterial suspensions was adjusted to an OD_600_ of 0.8, which is approximately 3.6 × 10^8^ CFU/mL. The rice plants used in this experiment were grown under greenhouse conditions (average 30 °C during the day and 25 °C at night). Subsequently, the stems of rice plants in the vegetative stage were inoculated with bacterial suspensions using a syringe and grown for eight days under greenhouse conditions. Disease severity was observed in the inoculated areas. At the flowering stage, each bacterial suspension was prepared and adjusted to OD_600_ = 0.5 to confirm BPB disease, and it was cultivated under greenhouse conditions with high relative humidity (close to 100%). The rice panicles were inoculated by dipping them in 50 mL of bacterial suspension for 1 min. At 8 days post-inoculation (dpi), disease severity in rice panicles was evaluated using the following scale: 0, healthy panicles; 1, 0–20% diseased panicles; 2, 20–40% diseased panicles; 3, 40–60% diseased panicles; 4, 40–80% diseased panicles; and 5, 80–100% diseased panicles. Disease severity was calculated using the following formula: Disease severity = Σ (number of samples per rating × rating value)/total number of panicles. The disease severity scores for wild-type BGR1, which produces toxoflavin, in *txeA*-expressing rice plants were also calculated in the same manner as described above.

### 4.6. Construction of txeA-Expressing Rice Plants

*Agrobacterium tumefaciens* GV3101 was transformed with the recombinant plasmid, pCAMBIA3301::*txeA*. Construction of transgenic rice plants was achieved using AgroTXE (plant transformation method using *A. tumefaciens* GV3101). The plants that appeared to have undergone a normal growing process judged from their appearance were selected. T1 (PIPK-X) and T2 (PIPK-X-X) transgenic rice plants were established. Transgenic rice plants were selected on medium containing phosphinothricin (100 μg/mL). Plant tissue was ground to a powder in liquid nitrogen using a TissueLyser II (Qiagen, Hilden, Germany). Total RNA was extracted using the RNeasy Plant Mini Kit (Qiagen) following the manufacturer’s recommendations. cDNA was synthesized from 500 ng of total RNA with txeA_cDNA_R as a target-specific primer using a SuperiorScript III cDNA synthesis kit (Enzynomics, Daejeon, Republic of Korea). To confirm *txeA* gene expression in rice plants, RT-PCR was performed under the PCR conditions mentioned above, using the synthesized cDNA as a template and the primer sets txeA_RT_sp_dF and txeA_RT_sp_dR.

### 4.7. Statistical Analysis

All experiments, except for the comparative proteomics analysis, were conducted twice with at least three replicates. Analysis of variance was conducted using the generalized linear model procedure, and the means were compared using the least significant difference test at *p* < 0.001, according to Tukey’s honestly significant difference test. Statistics for all experiments were performed using the Sigma Plot statistical program.

## 5. Conclusions

In this study, we aimed to unravel the complex relationship between the virulence factor toxoflavin and *B. glumae* BGR1, a pathogen that causes BPB in rice. Our investigation showed a link between toxoflavin and virulence, involving the introduction of metagenome-derived toxoflavin-degrading enzymes, the generation of gene deletion mutants incapable of toxoflavin synthesis, and their impact on bacterial virulence and plant resistance. These findings suggest the possibility of identifying new virulence factors in addition to the major ones in *B. glumae* BGR1. Moreover, the introduction of functional proteins such as TxeA holds significant potential for the development of innovative disease control strategies in rice agriculture.

## Figures and Tables

**Figure 1 plants-12-03934-f001:**
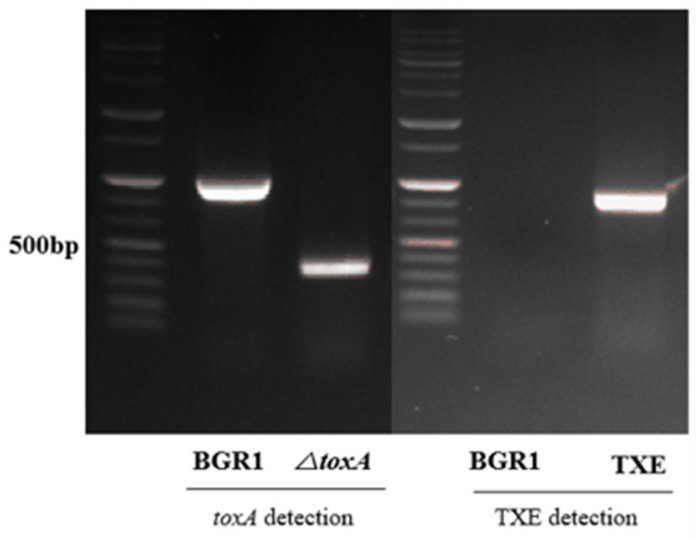
Confirmation of the mutant strains. The deletion mutant, *△toxA*, was generated and PCR was conducted using primers targeting the 5′-upstream and 3′-downstream regions of *toxA* to verify the mutant. Strains with successfully integrated *txeA*, called TXE, were generated, and PCR was conducted using cloning primers. Agarose gel electrophoresis of the PCR products was performed to distinguish *B. glumae* BGR1 and mutant strains based on different fragment lengths.

**Figure 2 plants-12-03934-f002:**
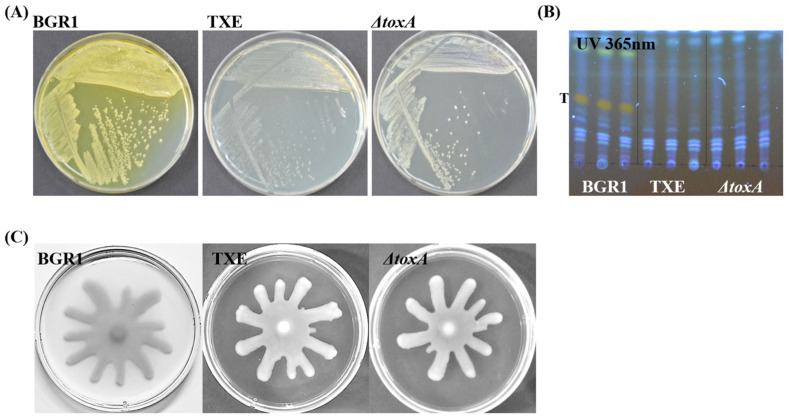
Phenotype assay of *Burkholderia glumae* BGR1 and mutant strains. (**A**) The pigmentation phenotype by toxoflavin was observed after three days of culture. (**B**) Toxoflavin production by *B. glumae* BGR1 and mutant strains was detected on thin-layer chromatography (TLC) silica gel plate. (**C**) Swarming motility by *B. glumae* BGR1 and mutant strains on 0.5% agar plates. These results are representative and from independent experiments replicated three times with the same patterns.

**Figure 3 plants-12-03934-f003:**
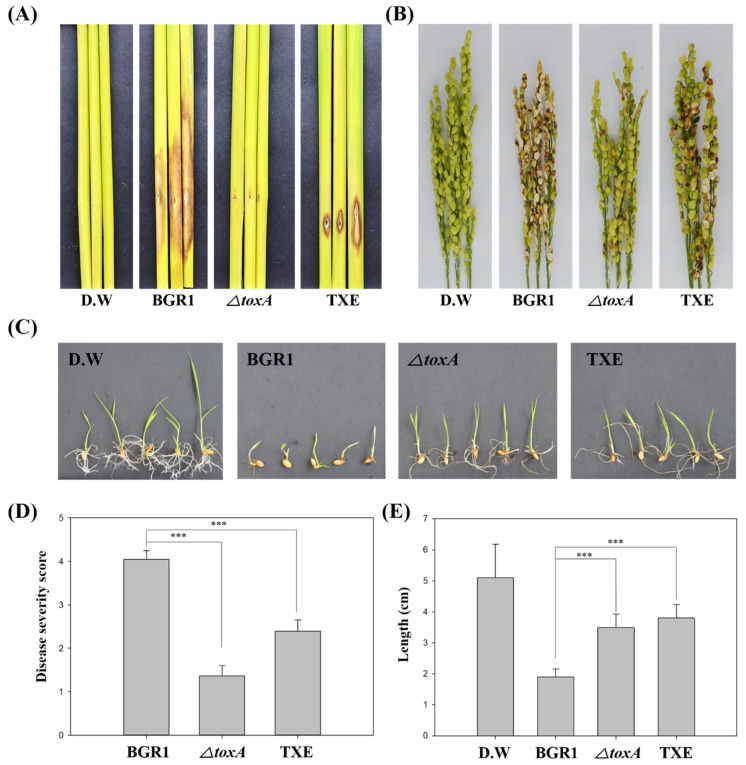
In vivo virulence assay at various plant stages to assess the virulence of toxoflavin-deficient mutant strains. (**A**) The virulence of toxoflavin at the vegetative stage. (**B**) The virulence of toxoflavin at the flowering stage. (**C**) The virulence of toxoflavin at the seedling stage. (**D**) Disease severity on the rice panicle was calculated on a scale of 0 to 5. (**E**) The length of rice shoots was measured at the same magnification. the results are representative of at least three replicates. Data are presented as the mean ± S.D. of at least three replicates (*n* ≥ 3). Mean values followed by the same letters are not significantly different according to Tukey’s honest significance test (***, *p* < 0.001). Distilled water (D.W) was used as negative control.

**Figure 4 plants-12-03934-f004:**
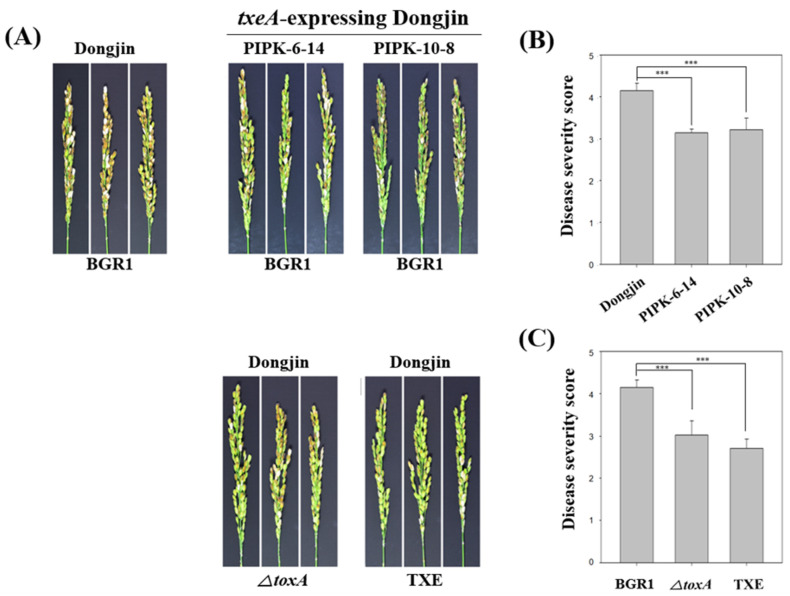
The correlation between toxoflavin and bacterial panicle blight as the virulence factor of *Burkholderia glumae* BGR1. (**A**) In vivo virulence analysis using *txeA*-expressing rice plants and non- toxoflavin non-producing mutant strains. (**B**) Calculation of disease severity score of *txeA*-expressing rice plants against *B. glumae* BGR1. (**C**) Calculation of disease severity score by mutant strains that do not produce toxoflavin. Data are presented as the mean ± S.D. of at least three replicates (*n* = 3). Mean values followed by the same letters are not significantly different according to Tukey’s honestly significant difference test (***, *p* < 0.001).

**Table 1 plants-12-03934-t001:** Bacterial strains and plasmids used in this study.

Name	Characteristics	Source
Bacterial strains		
*B. glumae*		
BGR1	*Burkholderia glumae* isolate from rice, wild type, Rif^r^	[1]
*Δ* *toxA*	BGR1 derivative, deletion of 384 bp within *bglu_2g06400* (*toxA*)	This study
TXE	BGR1 containing pBBR1MCS2::*txeA*	This study
*A.tumefaciens*		
GV3101	*Agrobacterium tumefaciens*, wild type, Rif	This study
AgroTXE	GV3101 containing pBBR1MCS2::*txeA*	This study
*E. coli*		
*E. coli* DH5α λpir	F^−^ 80d*lacZ*ΔM15 (*lacZYA-argF*) U169 *recA1 endA1hsdR17* (rk-, mk+) *phoAsupE44* -*thi-1 gyrA96 relA1*	Lab collection
*E. coli* S17-1 λpir	*hsdR recA* pro RP4-2 (Tc::Mu; Km::Tn7) (*λ pir*)	[22]
Plasmids		
pK18*mobsacB*	Allelic exchange suicide vector, *sacB* Km^r^	[23]
pK18*toxA*	For constructing *toxA* KO mutant, pK18*mobsacB*:: 448bp upstream-downstream of *toxA* region restricted by EcoRI-HindIII, respectively.	This study
pBBR1MCS2	Broad-host-range plasmid, Km^r^, used to construct *txeA*-expressing strains.	[24]
pBBR1MCS2::*txeA*	For expressing the gene of toxoflavin degrading enzyme (TxeA) mutant strain, pBBR1MCS2::CDS of *toxA*	This study
pCAMBIA3301	Binary vector, Km^r^, PPT^r^, used to construct *txeA*-expressing transgenic plant	[25]
pCAMBIA3301::*txeA*	For expressing the gene of toxoflavin degrading enzyme (TxeA) transgenic plants, pCAMBIA3301::CDS of *txeA*, *txeA* gene driven by CaMV 35S promoter.	This study

Abbreviation: Rif^r^, rifampicin resistance; Km^r^, kanamycin resistance; PPT^r^, phosphinothricin resistance.

**Table 2 plants-12-03934-t002:** Oligonucleotide primers used in this study.

Name	Sequence (5′→3′)	Use
toxA_LF	*TTTGGATCCTTGGCCCATCGATAGTGATT	To amplify the L fragment of *toxA* (*bglu_2g06400*)
toxA_LR	CGCGCCGATAGATTTCAC	To amplify the L fragment of *toxA* (*bglu_2g06400*)
toxA_RF	TTTTCGGGCGTGAAAAGCCAGTTCAGCTTCTAC	To amplify the R fragment of *toxA* (*bglu_2g06400*)
toxA_RR	*TTTAAGCTTGATTTGCGCAGAGTCTGGA	To amplify the R fragment of *toxA* (*bglu_2g06400*)
toxA_UP_F	CGAGGGAACATAGTGGCATT	To confirm the disruption of *toxA* (*bglu_2g06400*)
toxA_DOWN_R	TCTCAGGTTCGGGAGATACG	To confirm the disruption of *toxA* (*bglu_2g06400*)
pk18_DOWN_R	GTG AAG CTA GCT TAT CGC CAT	To confirm the first crossover in the process of constructing deletion mutant.
pBBR_TxeA_cF	*AAGGTACCAATGAATCAGCCTCCTCCT	Amplifying the *txeA* gene to be cloned in pBBR1MCS2.
pBBR_TxeA_cR	*TTTAAGCTTTCAATAGATCTTCCAAAT	Amplifying the *txeA* gene to be cloned in pBBR1MCS2.
pBBR1MCS2_dR	GACTCACTATAGGGCGAATTG	To confirm transformation in the process of constructing inserted *txeA* strains
pCAM_TxeA_cF	*AAAGGATCCATGAATCAGCCTCCTCCTCC	Amplifying the *txeA* gene to be cloned in pCAMBIA3301.
pCAM_TxeA_cR	*AAAAAGAATTCATAGATCTTCCAAATCCCAG	Amplifying the *txeA* gene to be cloned in pCAMBIA3301.
txeA_RT_sp_dF	GCGCATTTTAGAAACTTGTC	RT-PCR analysis to determine the *txeA* expression level
txeA_RT_sp_dR	TGAGGTGGGGATAGTTCCAAAC	RT-PCR analysis to determine the *txeA* expression level
txeA_cDNA_R	AGATCTTCCAAATCCCAGG	To construct the cDNA of *txeA*

* Underlined sequence in the primers indicates restriction enzyme targeted sequences.

## Data Availability

All data is comprised in the manuscript.

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
