# Peer review of "Understanding Burkholderia glumae BGR1 Virulence through the Application of Toxoflavin-Degrading Enzyme, TxeA"

_plants, 2023, doi:10.3390/plants12233934_

Round 1
Reviewer 1 Report
Comments and Suggestions for Authors
Bacterial panicle blight in rice, caused by Burkholderia glumae, is an economically important disease worldwide. Among various strategies, such as culture practices, chemical control, or biological control employed to control the disease, the use of resistant cultivars carrying factors resistant to this pathogen becoming a more sustainable and safe way for rice production. This paper provides some understanding of the relationship between the flavotoxin and virulence of Burkholderia glumae. The study also paved a promising strategy that uses a flavotoxin degradation enzyme to artificially create a novel resistant trait in rice against the pathogen.
I recommend accepting this paper in its present form.
Reviewer 2 Report
Comments and Suggestions for Authors
Kim and colleagues provide a study on the transformation of Burkholderia glumae BGR1 with the potential toxoflavin-degrading enzyme, TxeA and the generation of TxeA-expressing rice plants and its potential use.
The manuscript is of generell interest, but some issues have to be solved before it can be considered for publication.
There are two major flaws of the manuscript. First, the structure need some revision. Second, there is no empty vector control for the transformed plants.
Results:
Figure 3A: An empty vector control would have been an appropiate control.
Figure 4 should go to the Results part. Were do the PIPK-6-14 and PIPK-10-8 origin? There is no information in the Materials and Methods part.
Materials and Methods:
4.5. Pls add more detailed information on the plant growth. Which substrate, fertilzer, single pots, humidity in the greenhouse. Three replicates is a quite low number for plant experiments. How many CFU per ml were injected?
4.6. Why are there no empty vector control plants to exclude transformation effects? Which promotor was used to express the gene? This section should be more elaborated and go before 4.5.
4.7. Which programme was used for statistical analysis?
Delete 6. Patents from the manuscript, also line 399-408. Revise the Reference section carefully according to the journals standards.
Comments on the Quality of English LanguageModerate editing of English language required.
Round 2
Reviewer 2 Report
Comments and Suggestions for Authors
Please increase the quality of Figure 3. Especially numbers in D and E are too small to read.
Comments on the Quality of English LanguageEnglish proofreading is required.
